# Exploring Potential Mechanisms of Fludioxonil Resistance in *Fusarium oxysporum* f. sp. *melonis*

**DOI:** 10.3390/jof8080839

**Published:** 2022-08-11

**Authors:** Yan-Fen Wang, Fang-Min Hao, Huan-Huan Zhou, Jiang-Bo Chen, Hai-Chuan Su, Fang Yang, Yuan-Yuan Cai, Guan-Long Li, Meng Zhang, Feng Zhou

**Affiliations:** 1School of Resources and Environment, Henan Institute of Science and Technology, Xinxiang 453003, China; 2College of Plant Protection, Henan Agricultural University, Zhengzhou 450002, China; 3Institute of Vegetables and the Key Lab of Cucurbitaceous Vegetables Breeding in Ningbo City, Ningbo Academy of Agricultural Sciences, Ningbo 315040, China

**Keywords:** *Fusarium oxysporum* f. sp. *melonis*, fludioxonil, fungicide resistance, resistance mechanism, cross-resistance

## Abstract

Melon Fusarium wilt (MFW), which is caused by *Fusarium oxysporum* f. sp. *melonis* (FOM), is a soil-borne disease that commonly impacts melon cultivation worldwide. In the absence of any disease-resistant melon cultivars, the control of MFW relies heavily on the application of chemical fungicides. Fludioxonil, a phenylpyrrole fungicide, has been shown to have broad-spectrum activity against many crop pathogens. Sensitivity analysis experiments suggest that fludioxonil has a strong inhibitory effect on the mycelial growth of FOM isolates. Five fludioxonil-resistant FOM mutants were successfully generated by repeated exposure to fludioxonil under laboratory conditions. Although the mutants exhibited significantly reduced mycelial growth in the presence of the fungicide, there initially appeared to be little fitness cost, with no significant difference (*p* < 0.05) in the growth rates of the mutants and wild-type isolates. However, further investigation revealed that the sporulation of the fludioxonil-resistant mutants was affected, and mutants exhibited significantly (*p* < 0.05) reduced growth rates in response to KCl, NaCl, glucose, and mannitol. Meanwhile, molecular analysis of the mutants strongly suggested that the observed fludioxonil resistance was related to changes in the sequence and expression of the *FoOs1* gene. In addition, the current study found no evidence of cross-resistance between fludioxonil and any of the other fungicides tested. These results indicate that fludioxonil has great potential as an alternative method of control for FOM in melon crops.

## 1. Introduction

Melon (*Cucumis melo*), of which common varieties include musk melon, cantaloupe, and honeydew, is a popular fruit crop in the Cucurbitaceae family that is cultivated throughout the tropics and subtropics [1,2]. In addition to being a favorite for consumption due to their flavor, melons have also been noted for their nutritional and medicinal properties, some of which have been investigated as possible treatments for kidney stones, cancer, cardiovascular disorders, and stroke [3,4]. Although the data is incomplete, it has been estimated that over 30 million tons of melon is produced globally each year [5], while a recent report by the Food and Agriculture Organization of the United Nations (FAO) suggested that the total area under melon cultivation in China comprises as much as 427.8 thousand hectares, with a total output of 144 thousand tons (https://faostat3.fao.org/download/Q/QC/E, accessed on 16 January 2022). However, diseases have always posed a threat to successful melon production, especially melon Fusarium wilt (MFW) caused by *Fusarium oxysporum* f. sp. *melonis* (FOM) [6]. The continuous production of melons in monoculture in recent years has led to an increase in the incidence and severity of soilborne fungal diseases [7]. The spread of MFW is of particular concern as it can affect the entire growth cycle of the melon plant, especially from anthesis through to fruiting, when it can cause yield losses of 50–70% [8]. Although the development of disease-resistant cultivars would be the best option for the control of MFW [9], at present, there are no disease-resistant field cultivars available. Consequently the management of MFW relies primarily on cultural practices and the application of chemical fungicides, such as cupric ammonium complexion by root irrigation [9,10]. However, the inappropriate use of chemical fungicides has caused a series of problems, including the production of pesticide residues and environmental pollution, as well as the emergence of fungicide-resistant pathogens [11,12].

Fludioxonil, a phenylpyrrole fungicide, has been shown to have broad-spectrum activity against a wide range of fungal pathogens, and has been used to control many diseases in the field [13,14,15,16,17,18]. Previous studies have suggested that the mode action of fludioxonil is to inhibit fungal growth by overstimulating the high osmolarity glycerol (HOG) stress response signal transduction pathway, which causes the hyphae to swell and burst [19]. The HOG stress response pathway, a branched mitogen-activated protein kinase (MAPK) signal transduction system, has been well characterized in *Saccharomyces cerevisiae*, and is known to mediate osmoregulation in response to environmental factors [20,21]. The HOG1 protein itself is a key enzyme in the HOG1–MAPK signaling cascade [22], and has also been shown to play a role in the initial response to osmotic stress, prior to the activation of the HOG1–MAPK signaling pathway [23]. Although the molecular details are yet to be characterized fully, it is currently thought that the fludioxonil causes the hyperactivation of the HOG1–MAPK signaling transduction pathway by targeting group III hybrid histidine kinases (HK) such as *FoOs1* [13,14,15]. In addition to osmoregulation, the HOG1–MAPK pathway is involved in many fungal processes, including responding to heavy metal and oxidative stress, which perhaps explains why fludioxonil exhibits such a broad spectrum of activity against a wide range of fungal pathogens [13,15,24]. However, the inappropriate and excessive use of fludioxonil has already resulted in the emergence of fludioxonil resistance in many plant pathogens, including *Neurospora crassa*, *Alternaria brassicicola*, *Penicillium expansum*, *Sclerotinia sclerotiorum*, *Fusarium* spp., *Aspergillus carbonarius*, and *Botrytis cinerea* [17,25,26,27,28,29,30,31,32]. However, most previous studies have shown that the primary resistance mechanism appears to be associated with a set of well-characterized amino acid changes in the histidine kinase FoOs1 [25,26,32]. There have also been several reports of fludioxonil-resistant mutants that have no such mutations [33], indicating that the resistance mechanism of fludioxonil is more complicated than initially thought, and could involve multiple biological processes.

To date, the use of fludioxonil for the control of MFW has yet to be approved in China (http://www.icama.org.cn/hysj/index.jhtml, accessed on 16 January 2022). However, preliminary experiments conducted in the current study suggested that fludioxonil had a strong inhibitory effect on the mycelial growth of FOM isolates collected from melon-growing regions located in the Zhejiang province of China, even at low concentrations (Appendix A). Further research is therefore required to investigate the biological characteristics and molecular mechanism of fludioxonil-resistance, especially the risk of emergence of highly resistant isolates, to determine whether fludioxonil should be registered for the control of MFW in China. The objectives of the current study were to: (1) compare the fitness parameters and physiological characteristics of sensitive and fludioxonil-resistant isolates of FOM; (2) explore potential mechanisms of fludioxonil resistance associated with the *Fo**Os1* gene; and (3) determine whether there are any patterns of cross-resistance between fludioxonil and other commonly used fungicides for the control of soil-borne crop diseases, including tebuconazole, prochloraz, fluazinam, carbendazim, pyraclostrobin, kresoxim-methyl, and difenoconazole.

## 2. Materials and Methods

### 2.1. Fungicides and Fungal Isolates

Fungicides, commonly used to control of soil-borne crop disease caused by *Fusarium* sp., were selected as candidates for fludioxonil sensitivity and cross-resistance research in this study. Technical grade sources of eight fungicides (Table 1) were used to prepare 10,000 μg/mL stock solutions, which were dissolved either in acetone for the hydrophobic fungicides, or 0.1 M hydrochloric acid (HCl) in the case of carbendazim. The stock solutions were stored at 4 °C.

The FOM isolates used in the current study were collected from root and leaf samples of *C. melo* plants exhibiting typical symptoms of MFW growing in the melon fields of Taizhou and Ningbo in the Zhejiang province of China during the 2019 growing season. The *Fusarium* isolates were isolated and purified by conventional culture methods, and 11 isolates were selected on the basis of the sequences of their Internally Transcribed Spacer (ITS) and Translation Elongation Factor 1α (TEF-1α), as well as pathogenicity experiments to confirm the severity of wilt symptoms in the melon host [8]. The fungal cultures were incubated at 24 °C with a 12 h light/dark photoperiod, and preserved on potato dextrose agar (PDA) medium at 4 °C. Mycelial samples were prepared using the protocol of a previous study [18].

### 2.2. Generation of Fludioxonil-Resistant Mutants

Five wild-type FOM isolates (TY-1, TY-3, TY-5, TY-8, and TG-5) were selected for generation of fludioxonil-resistant mutants (TY-1-fR, TY-3-fR, TY-5-fR, TY-8-fR, and TG-5-fR) according to Zhou et al. [16,17,18]. Briefly, each mutant was subjected to 10 successive rounds of subculture on fludioxonil-free PDA before their EC_50_ values were re-evaluated on PDA containing fludioxonil at the following concentrations: 0, 1.0, 3.0, 9.0, 27.0, 81.0, 243.0, 729.0, and 1000.0 μg/mL. After incubation at 24 °C with a 12 h photoperiod for 4 days, the diameter of each colony was measured twice at right angles. The stability of the resistance was then estimated from the change in EC_50_ values calculated for the 1st, 4th, 6th, 8th, and 10th subculture. Each mutant was represented by three separate Petri dishes, and the entire experiment was performed in triplicate. Solvents were also tested on PDA medium to evaluate the effects on radial growth of FOM isolates.

### 2.3. Biological Characteristics of Fludioxonil-Resistant Mutants

#### 2.3.1. Growth and Sporulation

Five laboratory fludioxonil-resistant mutants (TY-1-fR, TY-3-fR, TY-5-fR, TY-8-fR, and TG-5-fR) and their parental isolates (TY-1, TY-3, TY-5, TY-8, and TG-5) were used. The mycelial growth was compared using the protocols of previous studies [18]. Briefly, mycelial plugs (5 mm) were taken from the edge of 3-day-old colonies and transferred to fresh PDA. The dishes were incubated at 24 °C, and the resulting colonies observed daily; the diameter of each was measured at 24, 48, 72, 96, and 120 h post inoculation (hpi). Each isolate was represented by five separate dishes, and the entire experiment performed in triplicate. Meanwhile, the sporulation of the five mutants and their parental isolates was assessed as macroconidia produced using the method detailed in a previous study [18], with modifications. Briefly, mycelial plugs (5 mm) were taken from the edge of 3-day-old colonies and transferred to fresh PDB medium, after incubation at 24 °C with shaking (120 rpm) for 4 days, the resulting macroconidia were harvested and counted using a hemocytometer (Shanghai Qiujing Biochemical Reagent Instrument Co., Ltd., Shanghai, China) after filtration with a double-wrapped sterile gauze. Each isolate was represented by five separate flasks, and the entire experiment performed twice. Fisher’s least significant difference test in the SPSS software (ver. 17.0; SPSS Inc., Chicago, IL, USA) was calculated to determine statistical significance among different mutants (or isolates).

#### 2.3.2. Cross-Resistance against Fungicides

Five laboratory fludioxonil-resistant mutants (TY-1-fR, TY-3-fR, TY-5-fR, TY-8-fR, and TG-5-fR) and their parental fludioxonil-sensitive isolates (TY-1, TY-3, TY-5, TY-8, and TG-5) were used to investigate the potential cross-resistance between fludioxonil and a range of other fungicides (Table 1) commonly used for the control of soil-borne crop diseases. The mycelial growth assay on the PDA medium described above was similarly applied.

Prochloraz, and fludioxonil on fludioxonil-sensitive isolates, were assayed at 0, 0.00625, 0.0125, 0.025, 0.05, 0.1, 0.2, 0.4, 0.8, 1.6, and 3.2 μg/mL of a.i. tebuconazole, fluazinam, carbendazim, pyraclostrobin, kresoxim-methyl, and difenoconazole were tested at 0, 0.05, 0.1, 0.2, 0.4, 0.8, 1.6, 3.2, and 6.4 μg/mL of a.i. fludioxonil on the fludioxonil-resistant mutants was assessed at 0, 1.0, 3.0, 9.0, 27.0, 81.0, 243.0, 729.0, and 1000.0 μg of a.i. per mL. For each treatment/isolate combination, three separate Petri dishes were employed, and the entire experiment was performed in triplicate. After four days of incubation at 24 °C with a 12 h photo-period, the diameter of each colony was measured twice at right angles, and the EC_50_ values were calculated according to Zhou et al. [17].

#### 2.3.3. Sensitivity to Osmotic Stress

The response of isolates TY-1-fR, TY-3-fR, TY-5-fR, TY-8-fR, TG-5-fR, TY-1, TY-3, TY-5, TY-8, and TG-5 to osmotic stress was assessed using the methods described in previous studies [18,34], with few modifications. Briefly, mycelial plugs (5 mm) from 3-day-old PDA cultures of each isolate were transferred onto fresh PDA amended with either KCl, NaCl, glucose, or mannitol at final concentrations of 0.5 or 1 M to induce osmotic stress. The diameters of the resulting colonies were measured after 72 h incubation at 24 °C with a 12 h photoperiod, and the percentage inhibition of mycelial growth calculated according to the formula detailed in previous studies [18,34]. Each treatment was represented by six replicate Petri dishes, and the entire experiment performed in triplicate. Fisher’s least significant difference test in the SPSS software (ver. 17.0; SPSS Inc.) was calculated to determine statistical significance among different mutants (or isolates).

#### 2.3.4. Cloning and Sequencing Analysis of the *FoOs1* Gene

For genomic DNA extraction, five laboratory fludioxonil-resistant mutants (TY-1-fR, TY-3-fR, TY-5-fR, TY-8-fR, and TG-5-fR) and their respective sensitive parental isolates (TY-1, TY-3, TY-5, TY-8, and TG-5) were cultured in Potato Dextrose Broth (PDB; Beijing Aoboxing Bio-tech Co., Ltd., Beijing, China) at 24 °C with shaking (130 rpm) for 72 h in accordance with the protocols of previous studies [18,34]. The mycelium was harvested using sterile filter paper, washed in sterile water, and flash frozen in liquid nitrogen. The total genomic DNA was then extracted using the Omega bio-tek Fungal DNA Kit (Omega bio-tek lnc., Guangzhou, China) in accordance with the protocol of the manufacturer. The resulting DNA was then used as a template to amplify the full-length sequence of the *FoOs1* gene (FRV6_07880) using the FoOs1-F1/FoOs1-R1 primer set (Table 2) designed with primer premier software (ver. 6.0., PREMIER Biosoft, Quebec, Canada). The polymerase chain reactions (PCR) themselves, as well as the cloning and sequence analysis, were conducted using the protocol of a previous study [18]. Sequences from the fludioxonil resistant mutants, and those of the sensitive parental sequences, were compared using the software DNAMAN (Lynnon Biosoft Inc., San Ramon, CA, USA).

#### 2.3.5. Relative Expression of *FoOs1* Gene

*FoOs1* relative expression levels of the five laboratory fludioxonil-resistant mutants (TY-1-fR, TY-3-fR, TY-5-fR, TY-8-fR, and TG-5-fR) and their respective sensitive parental isolates (TY-1, TY-3, TY-5, TY-8, and TG-5) were assayed by quantitative real-time PCR (qPCR). Total RNA was extracted from mycelial samples and prepared as previously described, using a fungal RNA kit (Omega bio-tek, Darmstadt, Germany) in accordance with the manufacturer’s instructions for use in qPCR analysis. First-strand cDNA was prepared with a PrimeScript RT reagent kit (TaKaRa, Dalian, China), and used as a template for the amplification of a 200 bp fragment of the *FoOs1* gene with the RT-FoOs1-F1/ RT-FoOs1-R1 primer set. The qPCR amplification was performed using the applied biosystems QuantStudio 6 Flex PCR detection system (Thermo Fisher, Carlsbad, CA, USA) in conjunction with SYBR Green I fluorescent dye (TaKaRa, Dalian, China). The relative expression of the *FoOs1* gene in comparison with the *F. oxysporum* TEF-1α reference gene, which was amplified with the RT- EF1α-F/ RT- EF1α-R primer set (Table 2), was determined according to method described in previous study [35]. The following amplification program was run: an initial denaturation at 95 °C for 10 s; 40 cycles of 95 °C for 5 s, 60 °C for 20 s; and dissociation protocol as 95 °C for 10 s, 60 °C for 40 s and 95 °C for 15 s. Three biological replicates were assessed for each isolate and mutant, and the entire experimented conducted three times, with the resulting values being used to calculate the mean expression and standard error (SE). The relative expression of each gene was calculated with the 2^−ΔΔCT^ method [36], and the data collected in this study were calculated by analysis of variance using SPSS software (ver. 17.0; SPSS Inc., Chicago, IL, USA) by the methods of Fisher’s least significant difference test (*p* = 0.05).

## 3. Results

### 3.1. Generation of Fludioxonil-Resistant Mutants, Hereditable Stability, Mycelial Growth, and Sporulation of Fludioxonil-Resistant FOM Mutants

No effects on FOM radial mycelial growth were recorded by the used solvents at the range of tested concentrations. Meanwhile, repeated exposure to fludioxonil resulted in the generation of five highly resistant mutants, TY-1-fR, TY-3-fR, TY-5-fR, TY-8-fR, and TG-5-fR, which had EC_50_ values of 225.84, 204.07, 230.42, 264.22, and 232.96 μg/mL, respectively. After 10 successive rounds of subculture, it was found that the EC_50_ values of the five mutants had not changed significantly (Appendix A), indicating that the resistance was stable.

### 3.2. Biological Characteristics of Fludioxonil-Resistant Mutants

#### 3.2.1. Growth and Sporulation

Initial exploration of the biological characteristics of the resistant mutants revealed that their rate of mycelial growth was unaffected, and did not differ significantly compared to the sensitive parental isolates, whether at 12, 24, 48, 72, 96, or 120 hpi (Figure 1; Appendix A). However, further investigation indicated that several of the mutants had reduced fitness with regard to sporulation, with four of the resistant mutants (TY-1-fR, TY-3-fR, TY-5-fR, and TG-5-fR) exhibiting significantly reduced levels of sporulation (*p* < 0.05) compared to their parental isolates (Figure 2), while the fifth, TY-8-fR, actually had significantly increased levels of the macroconidia compared to its parental isolate, TY-8.

#### 3.2.2. Cross-Resistance against Fungicides

No evidence of cross-resistance between fludioxonil and any of the other fungicides tested, including tebuconazole, prochloraz, fluazinam, carbendazim, pyraclostrobin, kresoxim-methyl, and difenoconazole (Table 3). This result indicates that fludioxonil could provide an alternative method of MFW control in areas for which resistance has already emerged to other fungicides, and that alternate or combined application of these fungicides could be used to reduce the risk of resistance developing in fludioxonil itself.

#### 3.2.3. Sensitivity to Osmotic Stress

Osmotic sensitivity analysis showed that the fludioxonil-resistant mutants were also found to be more susceptible to osmotic stress, exhibiting significantly (*p* < 0.05) reduced mycelial growth rates on PDA amended with KCl (Figure 3A), NaCl (Figure 3B), glucose (Figure 3C), and mannitol (Figure 3D), both at the higher concentration of 1 M or the lower concentration of 0.5 M. Taken together, these results provide further confirmation of a fitness cost associated with fludioxonil molecular resistance, particularly with regard to their response to osmotic stress, which is consistent with the established fludioxonil mode of action, as well as known resistance mechanisms.

#### 3.2.4. Cloning and Sequence Analysis of the *FoOs1* Gene

Changes in the FoOs1 sequences of the fludioxonil-resistant mutants of FOM are presented in Table 4. For example, the fludioxonil-resistant mutants TY-1-fR, TY-3-fR, TY-5-fR, and TG-5-fR contained two mutations, and although one mutation, E702Q, was common in all four, the second differed, with TY-3-fR and TY-5-fR both having a mutation that resulted in the A896T substitution, while TY-1-fR and TG-5-fR had changes at S564P and N537T, respectively. The fifth mutant, TY-8-fR, was found to have three mutations, again the common mutation E702Q, as well as the N537T mutation found in TG-5-fR, and the R66A substitution that was not found in any of the other mutants. The five mutants were also found to contain a range of other nucleotide point mutations that did not affect the amino acid sequence of the predicted FoOs1 protein, including two in the intron region, G835A and C841T, and several synonymous mutations in the coding region, including A1294G, G1916C, T2015C, C2108T, A2471G, T3533C, G3745A, and G4063A.

#### 3.2.5. Relative Expression of the *FoOs1* Gene

The qPCR analysis conducted in the current study found that the *FoOs1* gene was differentially expressed in all of the fludioxonil-resistant mutants (Figure 4). For example, in the absence of fludioxonil, the expression of *FoOs1* was significantly down-regulated (*p* < 0.05) in four of the resistant mutants (TY-1fR, TY-3fR, TY-5fR, and TY-8fR) compared to the sensitive parental isolates (TY-1, TY-3, TY-5, and TY-8), while the expression of *FoOs1* was significantly up-regulated (*p* < 0.05) in the fifth mutant (TG-5fR). However, in the presence of fludioxonil (0.1 μg/mL), *FoOs1* expression was found to be significantly lower (*p* < 0.05) than that of the sensitive isolates in all of the mutants assessed, even though expression was slightly elevated compared to the control treatment in TY-1fR, TY-3fR, and TY-8fR, and slightly depressed in TY-5fR and TG-5fR (Figure 4). In contrast, the addition of fludioxonil had a dramatic effect on the wild-type isolates, significantly increasing *FoOs1* expression (*p* < 0.05) by more than 100% in some cases. Taken together, these results not only suggest that the *FoOs1* gene plays an important role in the response of wild-type FOM isolates to fludioxonil, but also that the depressed levels of *FoOs1* expression observed in the mutants could contribute to the fludioxonil resistance mechanism resulting in reduced sensitivity to the fungicide treatment.

## 4. Discussion

*Cucumis melo* is an important vine crop belonging to the Cucurbitaceae family, and is one of the most widely cultivated fruits globally [1,2]. In recent years, the MFW caused by FOM has become an increasing threat to melon production [6,9]. In the absence of MFW-resistant melon varieties, damage from MFW can have a significant impact on the yield and quality of melon fruits [5,9,10]. Furthermore, although chemical fungicides can provide effective control of *Fusarium* spp. pathogens, relatively few fungicides have high efficacy against MFW. Fludioxonil has been found to have broad-spectrum activity against both basidiomycete and ascomycete pathogens [13,37]. Although fludioxonil has not yet been registered for the control MFW in China (http://www.icama.org.cn/hysj/index.jhtml, accessed on 16 January 2022), our preliminary investigation found that fludioxonil exhibited high efficacy against FOM (Appendix A), exerting both preventative and curative activity. The current study assessed 11 FOM isolates collected from plants exhibiting typical symptoms of MFW growing in the melon fields of Taizhou and Ningbo in the Zhejiang province of China, and found an average EC_50_ value of 0.03 μg/mL.

Previous studies regarding the biological characteristics of fludioxonil-resistant mutants of plant pathogenic fungi have consistently found fitness costs associated with fludioxonil resistance [17,18,28]. For example, nearly all the reports investigating laboratory mutants of *S. sclerotiorum*, *B. cinerea*, and *F. graminearum* have documented reduced levels of growth, sporulation, and pathogenicity, as well as increased sensitivity to osmotic stress [17,18,28]. The current study found that although the fludioxonil-resistant mutants of FOM did not appear to exhibit impaired growth, their sporulation was significantly affected (*p* < 0.05), with four of the mutants (TY-1-fR, TY-3-fR, TY-5-fR, and TG-5-fR) having reduced levels of sporulation, while the fifth (TY-8-fR) had elevated levels compared to their respective parental isolates. Reproduction via spores is a common reproductive method in fungal pathogens, and sporulation is an essential phase in the life cycle of many plant pathogenic fungi [38]. In general, a reduction in sporulation is considered a factor that can limit the potential of a pathogen to reproduce and spread to new hosts; therefore, this fitness cost is associated with lower levels of risk according to the Fungicide Resistance Action Committee (https://www.frac.info/docs/default-source/publications/monographs/monograph-2.pdf, accessed on 10 January 2022), whilst the increased levels of sporulation observed in TY-8-fR would suggest increased risk.

Although there are currently few reports of fludioxonil-resistant mutants emerging in crop production in the field [16,17,18,37], there have been numerous studies of laboratory mutants that can easily be induced through continual exposure to fludioxonil under laboratory conditions in species as diverse as *Cochliobolus heterostrophus*, *B. cinerea*, *Ustilago maydis*, *S. sclerotiorum*, *S. homoeocarpa*, and *F. graminearum* [16,17,18,28,32,39,40]. Many of these reports have provided clues as to the resistance mechanism, as changes in the expression and amino acid sequences of candidate target genes such as *FoOS1*, which is an important component of the HOG1–MAPK signaling pathway, are associated with fludioxonil resistance [26,32,41]. The full-length open reading frame (ORF) of the *FoOs1* gene from *F. oxysporum* was found to be highly homologous to the *FgOs1* gene, which is the potential action target gene of fludioxonil in *F. graminearum*, and maybe also related to fludioxonil resistance [18]. It is therefore of great interest that the current study also found similar changes associated with the *FoOs1* gene of the fludioxonil-resistant mutants of FOM. For example, four of the mutants (TY-1-fR, TY-3-fR, TY-5-fR, and TG-5-fR) exhibited two amino acid changes, including S564P and E702Q, E702Q and A896T, E702Q and A896T, and N537T and E702Q, while the fifth (TY-8-fR) had three: R66A, N537T, and E702Q. Furthermore, it is particularly noteworthy that the E702Q mutation was common to all of the resistant mutants, perhaps indicating a critical role of this substitution in the fludioxonil resistance of FOM. Further bioinformatics analysis revealed that the E702Q mutation was located between the HAMP domain (646–698) and the HisKA domain (714–777), which are located in the flexible region that is closely linked to the potential binding site of fludioxonil within the FoOs1 protein. Despite this, it is interesting to note that this particular amino acid mutation has not been documented in fludioxonil-resistant mutants of other plant pathogenic fungi. Indeed, none of the point mutations identified in the current study (Appendix A) have been documented in previous studies of other *Fusarium* species that have investigated the MAP kinase proteins of *F. asiaticum* and *F. graminearum* [18,34]. Further investigation, including verification through reverse genetics, is therefore required to determine the importance of the E702Q mutation in fludioxonil resistance in FOM, as well as its potential role in other plant pathogenic fungi.

Similarly, the qPCR analysis conducted in the current study found that the altered expression of the *FoOs1* gene might be linked to the observed fludioxonil resistance of the FOM mutants. In this case, all of the mutants exhibited altered *FoOs1* expression in the absence of fludioxonil, but perhaps of more interest was their relatively low *FoOs1* expression in the presence of fludioxonil compared to the dramatically increased expression observed in the wild-type isolates when exposed to fludioxonil. These results are consistent not only with the increased sensitivity of the fludioxonil-resistant mutants to osmotic stress, but also with the established fludioxonil mode of action that is thought to target the HOG1–MAPK signaling pathway, which is closely associated with osmoregulation in fungi. To date, it has generally been assumed that fludioxonil resistance is linked to amino acid changes in various MAPK proteins (Appendix A); however, the expression data from the current study and the discovery of at least one resistant mutant of *F. asiaticum* that lacks any mutations in its *FgOs1* gene [33,34] indicate that the fludioxonil resistance mechanism could be more complex than previously thought, and might result from multiple resistance mechanisms. Further investigation, including transcriptome, metabolome, and proteome analysis are therefore required to identify key genes and fully characterize the mechanisms by which fludioxonil resistance can emerge in plant pathogenic fungi. Such data would not only be extremely useful in managing the risk of fludioxonil resistance, but could also assist in the development of novel phenylpyrrole fungicides designed to target the HOG1–MAPK signaling pathway more effectively.

In conclusion, the results of the current study provide strong evidence that the use of fludioxonil could be an effective fungicide for the control of MFW, especially in areas where resistance to other fungicides has already emerged, while the fitness cost associated with fludioxonil resistance observed in the FOM mutants, with regard to both sporulation and increased osmotic sensitivity, could indicate a low to moderate risk of fungicide resistance emerging in the field. However, the lack of any cross-resistance between fludioxonil and other fungicides, such as tebuconazole, prochloraz, fluazinam, carbendazim, pyraclostrobin, kresoxim-methyl, and difenoconazole (Table 3), and natural biopesticides [42] could mitigate such risk if fludioxonil were applied alternately or in combination with these other compounds. Under such circumstances, it is likely that the appropriate use of fludioxonil within an integrated pest management (IPM) program could provide effective control of MFW and safeguard the future of melon production in China and worldwide.

## Figures and Tables

**Figure 1 jof-08-00839-f001:**
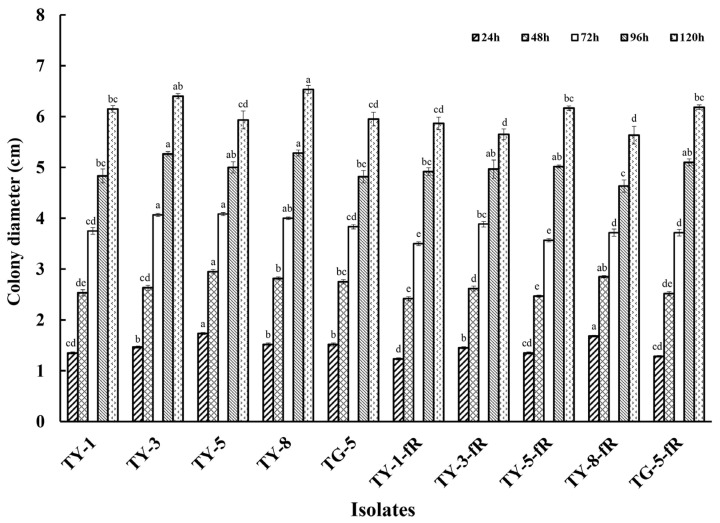
Mycelial growth of five fludioxonil-resistant mutants (TY-1-fR, TY-3-fR, TY-5-fR, TY-8-fR, and TG-5-fR) of FOM in comparison to their sensitive parental isolates (TY-1, TY-3, TY-5, TY-8, and TG-5) when grown on PDA medium at 24 °C. Data are the means of 10 replicates ± SE. Different letters above columns indicate significant differences according to Fisher’s least significant difference test (*p* ≤ 0.5).

**Figure 2 jof-08-00839-f002:**
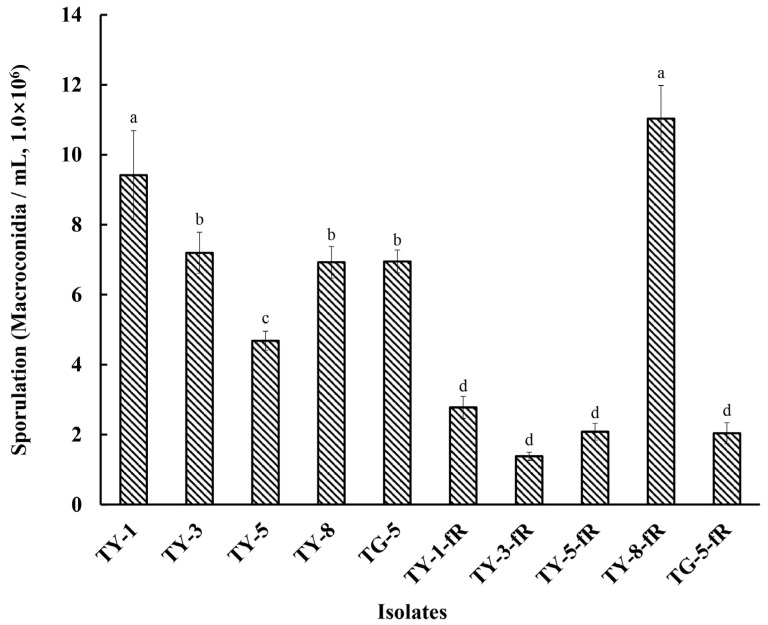
Sporulation of five fludioxonil-resistant mutants (TY-1-fR, TY-3-fR, TY-5-fR, TY-8-fR, and TG-5-fR) of FOM in comparison to their parental isolates (TY-1, TY-3, TY-5, TY-8, and TG-5). Data are the means of six replicates ± SE. Different letters above columns indicate significant differences according to Fisher’s least significant difference test (*p* ≤ 0.05).

**Figure 3 jof-08-00839-f003:**
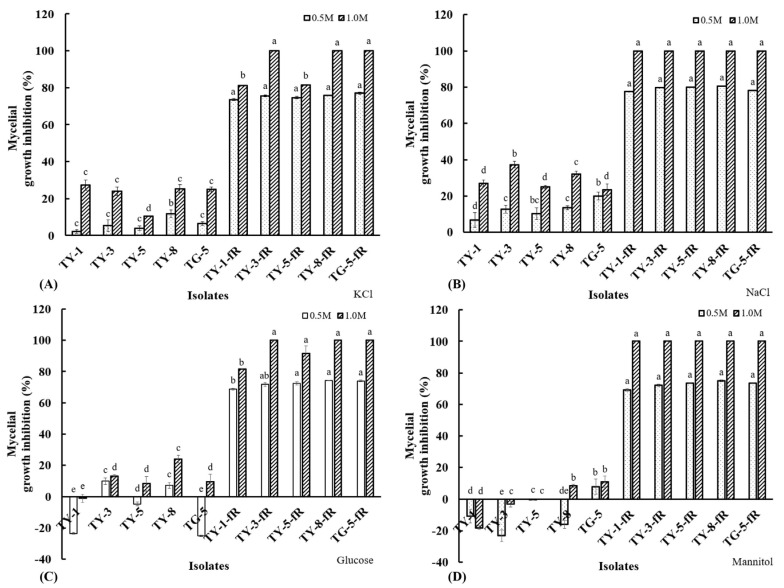
Mycelial growth inhibition of five fludioxonil-resistant mutants (TY-1-fR, TY-3-fR, TY-5-fR, TY-8-fR, and TG-5-fR) of FOM and their sensitive parental isolates (TY-1, TY-3, TY-5, TY-8, and TG-5) in response to osmotic stress induced by (**A**) KCl, (**B**) NaCl, (**C**) Glucose, (**D**) Mannitol at final concentrations of either 0.5 or 1 M. Data are the means of 12 replicates ± SE. Error bars indicate one standard error of the mean (SE) as calculated from the mean of two separate experiments, while different letters above columns indicate significant differences according to Fisher’s least significant difference test (*p* ≤ 0.05).

**Figure 4 jof-08-00839-f004:**
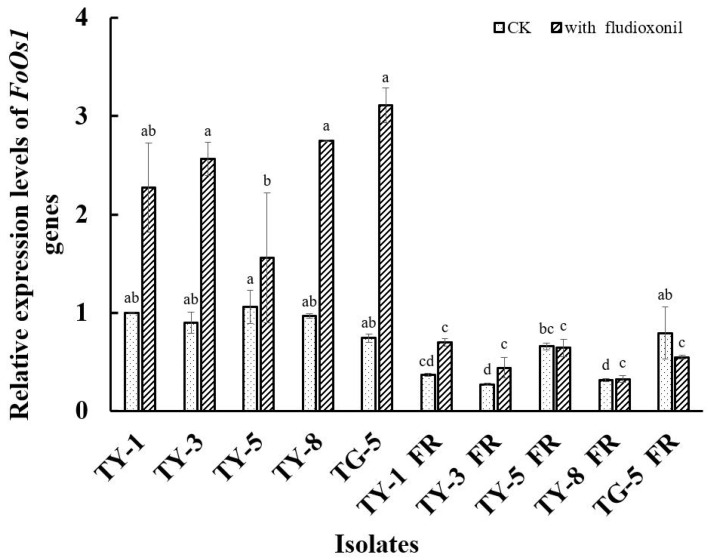
Relative expression of the *FoOs1* gene in five fludioxonil-resistant mutants (TY-1-fR, TY-3-fR, TY-5-fR, TY-8-fR, and TG-5-fR) of FOM and their sensitive parental isolates (TY-1, TY-3, TY-5, TY-8, and TG-5) in the absence and presence of fludioxonil. Data are the means of three replicates ± SE. Error bars indicate one standard error of the mean (SE) as calculated from the mean of two separate experiments, while different letters above columns indicate significant differences according to Fisher’s least significant difference test (*p* ≤ 0.05). CK indicates the expression level in the absence of fludioxonil.

**Table 1 jof-08-00839-t001:** Fungicides used in this study.

Active Ingredient (%)	Manufacturer	Fungicide Group	Mode of Action	FRAC * Code
Fludioxonil (96.0)	Hubei Jianyuan Chemical Co., Ltd. (Wuhan, China)	Phenylpyrrole	MAP/histidine kinase in osmotic signal transduction	12
Tebuconazole (96.2)	Sheyang Huanghai Pesticide Chemical Co., Ltd. (Yancheng, China)	DMI (14α-demethylase inhibitor)	Sterol biosynthesis in membranes	3
Prochloraz (97.0)	Hubei Kangbaotai Fine Chemical Co., Ltd. (Wuhan, China)	Sterol biosynthesis in membranes	C14-demethylase in sterol biosynthesis (erg11/cyp51)	3
Fluazinam (96.0)	Hubei Jianyuan Chemical Co., Ltd. (Wuhan, China)	Respiration	Uncouplers of oxidative phosphorylation	29
Pyraclostrobin(97.7)	Hubei Kangbaotai Fine Chemical Co., Ltd. (Wuhan, China)	Respiration	Complex III: cytochrome bc1 (ubiquinol oxidase) at Qo site (cyp b gene)	11
Kresoxim-methyl (97.5)	Jiangsu Gengyun Chemical Co., Ltd. (Zhenjiang, China)	Respiration	Complex III: cytochrome bc1 (ubiquinol oxidase) at Qo site (cyp b gene)	11
Difenoconazole (95.0)	Hubei Jianyuan Chemical Co., Ltd. (Wuhan, China)	DMI (14α-demethylase inhibitor)	Sterol biosynthesis in membranes	3
Carbendazim (98.1)	Haili Guixi Chemical Co., Ltd. (Yingtan, China)	Methyl benzimidazole carbamate	ß-tubulin assembly in mitosis	1

* FRAC = Fungicide Resistance Action Committee.

**Table 2 jof-08-00839-t002:** Primers used in this study for *FoOs1* full-length amplification (FLGA) and gene expression analysis (GEA).

Primers	Sequences (5′-3′)	Purpose	Sources
FoOs1-F1	ATGGTTGACGACGCGGCCCTCGCCGCT	Full-length gene amplification	Current study
FoOs1-R1	TTAGTTGGTAAGACTTCGCATATCAGAG		
RT-FoOs1-F1	GGCGTCAAATCTCACAGTCC	Gene expression analysis	[35]
RT-FoOs1-R1	AACTCGCTGCACTTCGTAAC		
RT-EF1α-F	CATCGGCCACGTCGACTCT		
RT-EF1α-R	AGAACCCAGGCGTACTTGAA		

**Table 3 jof-08-00839-t003:** Cross-resistance between fludioxonil and other commonly used fungicides.

Fungicides	Fludioxonil-Sensitive Isolates (EC_50_, μg/mL)					Fludioxonil-Resistant Mutants (EC_50_, μg/mL)				
	TY-1	TY-3	TY-5	TY-8	TG-5	TY-1-fR	TY-3-fR	TY-5-fR	TY-8-fR	TG-5-fR
Fludioxonil	0.03 *	0.02	0.03	0.03	0.04	225.84	204.07	230.42	264.22	232.96
Tebuconazole	0.33	0.23	0.22	0.38	0.31	0.39	0.32	0.34	0.43	0.40
Prochloraz	0.003	0.003	0.003	0.002	0.001	0.007	0.002	0.006	0.003	0.003
Fluzainam	0.051	0.033	0.04	0.041	0.02	0.049	0.035	0.053	0.016	0.058
Carbendazim	1.094	0.793	0.748	0.706	0.633	0.536	0.467	0.403	0.484	0.506
Pyraclostrobin	0.114	0.097	0.104	0.130	0.106	0.687	0.646	0.206	0.518	0.369
Kresoxim-methyl	0.392	0.581	0.528	0.520	0.119	0.569	0.295	0.599	0.238	0.764
Difenoconazole	0.303	0.212	0.256	0.207	0.167	0.222	0.119	0.232	0.119	0.800

* Values indicate the EC_50_ (50% effective concentration) for each isolate/fungicide combination.

**Table 4 jof-08-00839-t004:** Mutations in the predicted FoOs1 protein sequence of five fludioxonil-resistant mutants of FOM.

Mutants	Mutation Type and Location	
	Nucleotide	Amino Acid
TY-1-fR	A1294G *, T1863C **, G1916C, T2015C, C2108T, A2330G, A2471G, T3533C, G3745A, and G4063A	S564P and E702Q
TY-3-fR	A1294G, G1916C, C2108T, A2330G, A2471G, G2910A, T3533C, G3745A, and G4063A	E702Q and A896T
TY-5-fR	A1294G, G1916C, C2108T, A2330G, A2471G, G2910A, T3533C, G3745A, and G4063A	E702Q and A896T
TY-8-fR	T66C, A1722C, G1916C, C2108T, A2330G, A2471G, T3533C, G3745A, and G4063A	R66A, N537T, and E702Q
TG-5-fR	A1722C, G1916C, A2330G, T3533C, G3745A, and G4063A	N537T and E702Q

* Several of the nucleotide point mutations were found to have no effect on the amino acid sequence of the predicted protein, including G835A and C841T, which were located in the intron region, and A1294G, G1916C, T2015C, C2108T, A2471G, T3533C, G3745A, and G4063A, which were synonymous mutations. ** The remaining point mutation did cause amino acid changes, which are detailed as follows: T1863C (S564P), A2330G (E702Q), G2910A (A896T), T66C (R66A), and A1722C (N537T).

## Data Availability

Not applicable.

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
