# Peer review of "Exploring Potential Mechanisms of Fludioxonil Resistance in *Fusarium oxysporum* f. sp. *melonis"

_jof, 2022, doi:10.3390/jof8080839_

Round 1

Reviewer 1 Report

The manuscript of Wang et al., entitled  "Exploring potential mechanisms of fludioxonil resistance in Fusarium oxysporum on Melon" describes the analysis of a large number of F. oxysporum f.sp. melonis isolates for their sensitivity to fludioxonil. Furthermore mutants were generated and these were tested for mutations in the potential target gene of fludioxonil, and the effect on conditions regulated via the HOG1-MAPK pathway. 

In essence the manuscript is well written, with only a few sentences to be corrected (see below). My major remarks are that 1) the manuscript is highly repetitive with previous work from the same group on Fusarium graminearum (Zhou et al., 2020, Plant disease) and B. cineara (Zhou et al., 2020, Plant Disease) en therefore not novel. 2) Furthermore, trailing back on the data, can there be an explanation on the large number of mutations developed in just few generations? Would a genome wide sequencing reveal mutations over whole genome (coding/non-coding) or how is this achieved (only) in FoOs1.. Currently, I have no clue so please educate me on this.

Minor points.

-Lines 20-24, this sentence is incorrect and I think a part was deleted?
-Line 53, the end of this sentence is incorrect.
-Line 112, please indicate the composition of the plates (is this PDA?), also with the reference in 125 to this sentence, this would be neat to add.
-Line 145, I think this should be another reference (17 instead of 18..), although the latter did not describe a repetitive exposure so I am confused, maybe write "in short" the full protocol.
-Line 403, talking about a gene, should be in Italics
General, please arrange that Table's start on a page, not overlapping two of them.

Author Response

Reviewer 1:

The manuscript of Wang et al., entitled "Exploring potential mechanisms of fludioxonil resistance in Fusarium oxysporum on Melon" describes the analysis of a large number of F. oxysporum f.sp. melonis isolates for their sensitivity to fludioxonil. Furthermore, mutants were generated and these were tested for mutations in the potential target gene of fludioxonil, and the effect on conditions regulated via the HOG1-MAPK pathway. 

In essence the manuscript is well written, with only a few sentences to be corrected (see below). My major remarks are that 1) the manuscript is highly repetitive with previous work from the same group on Fusarium graminearum (Zhou et al., 2020, Plant disease) and B. cineara (Zhou et al., 2020, Plant Disease) en therefore not novel. 2) Furthermore, trailing back on the data, can there be an explanation on the large number of mutations developed in just few generations? Would a genome wide sequencing reveal mutations over whole genome (coding/non-coding) or how is this achieved (only) in FoOs1. Currently, I have no clue so please educate me on this.

Response: Thanks for your careful review and help suggestion. Fludioxonil, a phenylpyrrole fungicide, has been shown to have broad-spectrum activity against a wide range of fungal pathogens and has been used to control many diseases in the field. To date, the use of fludioxonil for the control of MFW has yet to be approved in China (http://www.icama.org.cn/hysj/index.jhtml). However, preliminary experiments conducted in the current study suggested that fludioxonil had a strong inhibitory effect on the mycelial growth of FOM isolates collected from melon-growing regions located in the Zhejiang province of China, even at low concentrations. Further research is therefore required to investigate the biological characteristics and molecular mechanism of fludioxonil-resistance, especially the risk of highly resistant isolates emerging, to determine whether fludioxonil should be registered for the control of MFW in China. These results provide data support for further understanding the molecular mechanism of resistance in different pathogenic fungi (including Fusarium graminearum, Botrytis cineara and Fusarium oxysporum) to fluromonil.

Minor points.

-Lines 20-24, this sentence is incorrect and I think a part was deleted?

Response: Thank you very much for you suggestion, we have revised the manuscript as suggested.

-Line 53, the end of this sentence is incorrect.

Response: We have revised the manuscript as suggested.

-Line 112, please indicate the composition of the plates (is this PDA?), also with the reference in 125 to this sentence, this would be neat to add.

Response: We have revised the manuscript as suggested.

-Line 145, I think this should be another reference (17 instead of 18..), although the latter did not describe a repetitive exposure so I am confused, maybe write "in short" the full protocol.

Response: We have revised the manuscript as suggested.

-Line 403, talking about a gene, should be in Italics.

Response: We have revised the manuscript as suggested.

General, please arrange that Table's start on a page, not overlapping two of them.

Response: We have revised the manuscript as suggested.

Reviewer 2 Report

Dear Authors,

The manuscript jof-1766206-peer-review-v1 “Exploring potential mechanisms of fludioxonil resistance in Fusarium oxysporum on Melon” presents interesting data on the resistance of Fusarium oxysporum f. sp. melonis against fludioxonil.

If I correctly read the paper, five fludioxonil-resistant Fusarium oxysporum f. sp. melonis mutants were obtained by repeated exposure to the chemical under laboratory conditions. The mutants significantly reduced mycelial growth and sporulation in the presence of the fungicide. Further investigation revealed that the fludioxonil-resistant mutants significantly reduced growth rates in response to osmotic stress caused by KCl, NaCl, glucose, and mannitol exposure. Molecular analysis suggested that the fludioxonil resistance was related to changes in the sequence and expression of the FoOs1 gene. Fludioxonil-resistant mutants maintain their sensibility to fungicides commonly used to control soil-borne fungal pathogens such as tebuconazole, prochloraz, fluazinam, carbendazim, pyraclostrobin, kresoxim-methyl and difenoconazole.

My general impression of this study is positive. Its presentation in the form of a manuscript is exposed in a cumbersome way and requires some adjustments.

Some suggestions to improve the manuscript.

Since Fusarium oxysporum comprises varieties and formae speciales, I suggest the use of Fusarium oxysporum f.sp. melonis (eventually abbreviate as FOM).

Usually, EC50 shows “50” as a subscript.

Verify the correct use of the International System of Units.

Insert the Authors name to all the species at the first citation (e.g., Fusarium oxysporum f. sp. melonis, Neurospora crassa, Alternaria brassicicola, Penicillium expansum, Sclerotinia sclerotiorum, Aspergillus carbonarius and Botrytis cinerea) or delete the Author name of Cucumis melo (lines 33, 116, and 340). Following the practice adopted by the International Code of Nomenclature for Algae, Fungi and Plants (ICN), in manuscripts dealing with taxonomy, for every organism, the full genus name and authority of the species should be included at first mention. For manuscripts dealing with subjects other than taxonomy, this is desirable, but not essential.

Title: use “f. sp. melonis” instead of “on Melon”

Enhance abstract content.

Keywords: use “Fusarium oxysporum f. sp. melonis” instead of “Fusarium oxysporum”

In the “Introduction” section a description of Melon Fusarium Wilt could help the readers.

Line 52: use “MFW” instead of “melon diseases”

Line 53: delete “was used”

Line 54: What does “chronic overuse” mean in this sentence?

Lines 83-91: This sentence is pertinent to results. Delete or re-write.

Materials and methods

I suggest the subsections:

            2.1. Fungicides and fungal isolates

            2.2. Generation of fludioxonil-resistant mutants of F. oxysporum f. sp. melonis

            2.3. Growth and sporulation of fludioxonil-resistant mutants

            2.4. Fungicides sensitivity

            2.5. Sensitivity to osmotic stress

            2.6. Cloning and sequencing analysis of the FoOs1 gene

            2.7. Relative expression of FoOs1 gene

Lines 101-111: I suggest the sentence “Technical grade sources of the fungicides listed in Table 1 were used in this study.”. Table 1 combines data from lines 101-111 and Table S1 in the following order: Active ingredient; Manufacturer; Fungicide group; Mode of action; FRAC code

Lines 112-114: delete “, and mycelial growth assays performed to confirm that the solvents had no effect on the growth of F. oxysporum at the range of concentrations tested (data not shown)”. This sentence is not pertinent to the section.

Line 121: What does “briefly” mean in this sentence?

Line 124: use “2.2. Fungicides sensitivity” instead of “2.2. Cross-resistance between fludioxonil and other commonly used fungicides”

Lines 125-138: describe the mycelial growth assay with particular attention to media, tested concentrations, growth conditions, etc. Indicate the strains used in this assay.

Lines 137-138: the paper Zhou et al. (2014) is in the text but is not present in the “References” section.

Lines 139-140: replace with “2.3. Generation of fludioxonil-resistant mutants of F. oxysporum f. sp. melonis

Lines 141-156: re-write. Avoid results information and maintain “Materials and Methods” information.

Line 157: delete “Mycelial”

Line 171: What does “double sterile gauze” mean in this sentence? Gauze (I suggest cheesecloth) was sterilized two times, or macroconidia were collected after filtration through two layers of sterile cheesecloth.

Line 175: delete “of fludioxonil-resistant mutants” The bioassay considers fludioxonil-resistant mutants and their parental strains.

Lines 191-194: delete “The … [18].” The sentence is not pertinent to “Materials and Methods” information.

Line 208: use “Primers used in this study for FoOs1 full-length amplification (FLGA) and gene expression analysis (GEA)

Table 1: use the above acronyms. FLGA instead of “Full-length gene amplification” and GEA instead of “Gene expression analysis”. Use [35] instead of “Karlsson et al., 2021”

Lines 210-211: use “2.7. Relative expression of FoOs1 gene” instead of “2.7. Determination … PCR”

Lines 212-213: use “FoOs1 relative expression levels of the five laboratory mutants and their respective sensitive parental strains were assayed by quantitative real-time PCR (qPCR).” instead of “The relative … (qPCR).”

Lines 214- 219: The sentence “Total RNA … primer set.” is not clear. Re-write.

Line 219: use “qPCR” instead of “real-time PCR”

Lines 219-224: the sentence “The real-time PCR amplification … previous study [35].” is not clear. Rewrite.

Line 225: use “The following amplification program was run:” instead of “Procedures … an”

Line 226: delete “following program by”

Lines 226-227: check and control the program steps.

Results

Avoid discussion information.

I suggest the subsections:

            3.1. Sensitivity of Fusarium oxysporum f.sp. melonis to fludioxonil

            3.2. Hereditable stability, mycelial growth, sporulation, resistance to other fungicides, and osmotic sensitivity

            3.3. Sequence analysis of the FoOs1 gene from fludioxonil-resistant mutants of Fusarium oxysporum f.sp. melonis

            3.4. Relative expression of FoOs1 gene

Lines 235-236: use “3.1. Sensitivity of Fusarium oxysporum f.sp. melonis to fludioxonil”

Line 239: insert “3.2. Hereditable stability, mycelial growth, sporulation, resistance to other fungicides, and osmotic sensitivity” between “respectively. ” and “ After” as a subsection.

Line 237: What does “generation” mean in this sentence?

Lines 237-239: improve.

Line 241: use “resistance was” instead of “mutations were genetically”

Lines 241-253: improve.

Figure 1: use “Colony diameter” instead of “Mycelial growth”

Line 255: insert “f.sp. melonis (TY-1-fR, TY-3-fR, TY-5-fR, TY-8-fR, TG-5-fR)” between “oxysporum ” and “in comparison”

Line 256: insert “ (TY-1, TY-3, TY-5, TY-8, TG-5)” between “isolates ” and “when”

Line 256-257: insert “Data are the means of ??? replicates ± SE.” between “24°C.” and “Different”

Figure 2: use “Macroconidia” instead of “Spores”

Line 260: insert “f.sp. melonis (TY-1-fR, TY-3-fR, TY-5-fR, TY-8-fR, TG-5-fR)” between “oxysporum ” and “in comparison”

Line 261: insert “ (TY-1, TY-3, TY-5, TY-8, TG-5). Data are the means of ??? replicates ± SE.” between “isolates ” and “. Different”

Figure 3:

Use “Radial growth inhibition (%)” instead of “Percentage of mycelial growth inhibition (%)”

Explain red numbers. I suggest the use of “-20” and “-40” instead of “20” and “40”, respectively

If it is possible delete “120”.

There are two sections “A” and two “D”. I recommend inserting the stress agent in the graph, at the top left, near the ordinate axis; delete “A”, “B”, “C” and “D” in the plots and “(A)”, “(B)”, “(C)” and “(D)” in the legend.

Line 272: insert “ (TY-1-fR, TY-3-fR, TY-5-fR, TY-8-fR, TG-5-fR)” between “mutants” and “of”

Line 272: use “oxysporum f.sp. melonis” instead “oxysporum”

Line 273: insert “ (TY-1, TY-3, TY-5, TY-8, TG-5)” between “isolates” and “in”

Lines 274-275: use “Data are the mean of two separate experiments each with ??? replicates. Different” instead of “Error bars … while different”

Lines 278-295: re-write deleting information regarding the discussion. Line 256: insert “ (TY-1, TY-3, TY-5, TY-8, TG-5)” between “isolates ” and “when”

Line 297: use “oxysporum f.sp. melonis” instead “oxysporum”

Table 2 and related notes: the presence of “A” and “B” as upper scripts is confusing. I propose symbols like “*” and “**” or similar.

Lines 304-321: improve.

Lines 322-328: see figures 1 and 2 indications.

Lines 339-435: This section is also particularly difficult to read. Different concepts are repeated and redundant. Improve.

Adjust the references following the journal guidelines.

Figure S1: insert a legend.

Figure S2: insert a legend.

Reviewer 3 Report

Dear authors

Please revise your manuscript acoording to my comments in the attached file pdf,  and also cite the following paper in your introduction part related the use of natural biopesticdes as possible alternative to chemical fungicides.

Camele I., Elshafie H.S., Caputo L., Sakr S.H. and De Feo V.  2019. Bacillus mojavensis: Biofilm formation and biochemical investigation of its bioactive metabolites. J. Biol. Res. 92: (8296), 39-45. DOI: 10.4081/jbr.2019.8296.

Best regards

Author Response

Reviewer 3:

Please revise your manuscript acoording to my comments in the attached file pdf, and also cite the following paper in your introduction part related the use of natural biopesticdes as possible alternative to chemical fungicides.

Camele I., Elshafie H.S., Caputo L., Sakr S.H. and De Feo V.  2019. Bacillus mojavensis: Biofilm formation and biochemical investigation of its bioactive metabolites. J. Biol. Res. 92: (8296), 39-45. DOI: 10.4081/jbr.2019.8296.

Best regards

Response: Thanks for your careful review and help suggestion, we have revised the manuscript as your suggested. Meanwhile, these comments are all valuable and very helpful for revising and improving our manuscript, and some other details in the paper have also been revised in detail.

Round 2

Reviewer 2 Report

Dear Authors,

Unfortunately, the indications given with the first revision have been partly disregarded.

In some parts the manuscript is still very confusing.

The authors did not take care of the layout of the manuscript that presents figures and legend on different pages.

I consider the manuscript still invalid for publication in the Journal of Fungi.

Enhance abstract content.

Line 17: What does “Preliminary experiments” mean in this sentence?

Line 81: use “. There” instead of “, there”

Line 324: delete “L.”

“Material and Methods” section is difficult to understand.

Fungicides listed on lines 102-109 are the same present on table S1. For this reason, I suggest combining these data in a table labelled as “Table 1”.

Table 1. Fungicides used in this study

Active ingredient (grade, %)

Manufacturer

Fungicide group

Mode of action

FRACz code

Line 115: delete “initially”

In order to understand the lab steps, I suppose that:

1) FOM strains (11) were isolated during 2019 from Cucumis melo plants affected by MFW and identified according to ITS and TEF-1α sequences, as well as pathogenicity tests on melon host as reported by [8].

2) The fungal cultures were stored on PDA medium at 4°C, and mycelial samples prepared using the protocol of a previous study [18].

3) FOM strains were assayed for fludioxonil sensitivity

4) Five strains (TY-1, TY-3, TY-5, TY-8, and TG-5) were selected

5) The five strains were exposed to fludioxonil

6) Five highly resistant laboratory mutants were selected (TY-1-fR, TY-3-fR, TY-5-fR, TY-8-fR, and TG-5-fR)

7) The five highly resistant laboratory mutants (TY-1-fR, TY-3-fR, TY-5-fR, TY-8-fR, and TG-5-fR) and their parental isolates (TY-1, TY-3, TY-5, TY-8, and TG-5) were assayed for:

      7a) Sensitivity against tebuconazole, prochloraz, fluazinam, carbendazim, pyraclostrobin, kresoxim-methyl, and difenoconazole

      7b) Growth rate and sporulation

      7c) Sensitivity to osmotic stress

      7d) Cloning and sequencing analysis of the FoOs1 gene

      7e) Relative expression of FoOs1 gene

If these steps are correct, arrange materials and methods sections in the following subsections:

            2.1. Fungicides and fungal isolates

            2.2. Generation of fludioxonil-resistant mutants

            2.3. Biological characteristics of fludioxonil-resistant mutants

            2.3.a. Growth and sporulation

            2.3.b. Cross-resistance against fungicides

            2.3.c. Sensitivity to osmotic stress

            2.3.d. Cloning and sequencing analysis of the FoOs1 gene

            2.3.e. Relative expression of FoOs1 gene

Lines 112-114: Materials and methods could report: “Solvents were tested on PDA medium to evaluate the effects on radial growth of FOM strains”. The results could report “No effects on FOM radial mycelial growth were recorded by the used solvents at the range of tested concentrations (data not shown).”

Table 1: combines FoOs1-F1, FoOs1-R1, RT-FoOs1-F1 and RT-FoOs1-R1 under a unique “Current study” as “Sources”

Lines 205-211: the sentences “Total … primer 35].” Remain confused and not clear.

Results sections should be arranged following the suggested materials and method subsections.

Adjust the references following the journal guidelines. In particular: 5, 7, 17, 18, 34, and 42 (Camele…). Reference 41 is not present and n° 42 associates two papers.

Figure S1 and Figure S2 legends are not present.

Round 3

Reviewer 2 Report

This new revised version of the manuscript is still very confusing in some parts.

The authors did not take care of the layout of the manuscript that presents figures and legends on different pages and unformatted tables.

Enhance abstract and discussion content.

Usually, the gene names are in italics.

The “Material and Methods” section is difficult to understand and reports unnecessary results.

The FOM isolates growth conditions are not well explained: indicate the quality of light used (Radiant flux or intensity).

Line 18: What does “pydiflumetofen-resistant” mean in this sentence?

Line 34: delete “ L.”

Table 1:

1) format as table 3 on full page

2) enlarge the width of the first 4 columns.

3) Because line 103 indicates “Technical grade sources of eight fungicides (Table 1)”, indicate in the heading of column “Active ingredient (%)” and delete “Technical, %” in the row.

 Lines 106-107: delete “, and the solvents were tested on potato dextrose agar (PDA) medium to evaluate the effects on radial growth of FOM isolates”

Lines 121-127: use “Five wild-types FOM isolates (TY-1, TY-3, TY-5, TY-8, and TG-5) were selected for generation of fludioxonil-resistant mutants according to Zhou et al [16-18]. Briefly, each fludioxonil-resistant mutant” Instead of “The wild-type … mutant”

Line 134: add “Solvents were also tested on PDA medium to evaluate the effects on radial growth of FOM strains”

Line 136: delete “Mycelial”

Lines 137-139: use “Five laboratory fludioxonil-resistant mutants (TY-1-fR, TY-3-fR, TY-5-fR, TY-8-fR, and TG-5-fR) and their parental isolates (TY-1, TY-3, TY-5, TY- 138 8, and TG-5) were used. The mycelial growth was compared using the protocols of previous studies [18].” Instead of “ The mycelial growth … studies [18].”

Lines 144-146: use “Meanwhile, the sporulation of the five mutants and their parental isolates was assessed as macroconidia produced using the method detailed in a previous study [18], with modifications.” instead of “Meanwhile … modifications.”

Line 154: use “2.3.b. Cross-resistance against fungicides” instead of “.3.b. Cross-resistance between fludioxonil and other commonly used fungicides”

Lines 155-168 use: “Five laboratory fludioxonil-resistant mutants (TY-1-fR, TY-3-fR, TY-5-fR, TY- 137 8-fR, and TG-5-fR) and their parental Fludioxonil-sensitive isolates (TY-1, TY-3, TY-5, TY- 138 8, and TG-5) were used to investigate the potential cross-resistance between fludioxonil and a range of other fungicides (Table 1) commonly used for the control of soil-borne crop diseases. The mycelial growth assay on the PDA medium described above was similarly applied.

Prochloraz, and fludioxonil on fludioxonil-sensitive isolates were assayed at 0, 0.00625, 0.0125, 0.025, 0.05, 0.1, 0.2, 0.4, 0.8, 1.6, and 3.2 μg/mL of a.i. Tebuconazole, fluazinam, carbendazim, pyraclostrobin, kresoxim-methyl, and difenoconazole were tested at 0, 0.05, 0.1, 0.2, 0.4, 0.8, 1.6, 3.2, and 6.4 μg/mL of a.i. Fludioxonil on the fludioxonil-resistant mutants was assessed at 0, 1.0, 3.0, 9.0, 27.0, 81.0, 243.0, 729.0, and 1000.0 μg of a.i. per mL. For each treatment/isolate combination three separate Petri dishes were employed, and the entire experiment was performed in triplicate. After four days of incubation at 24°C with a 12 h photo-period, the diameter of each colony was measured twice at right angles, and the EC50 were calculated according to Zhou et al [17].” Instead of “The mycelial growth … et al [17].”

Lines 170-173 use “The response of strains TY-1-fR, TY-3-fR, TY-5-fR, TY-8-fR, TG-5-fR, TY-1, TY-3, TY-5, TY-8, and TG-5 to osmotic stress was assessed using the methods described in previous studies [18,34], with few modifications. Briefly, mycelial plugs (5 mm) from 3-day-old PDA cultures of each isolate were transferred onto fresh PDA amended with either KCl, NaCl, glucose, or Mannitol at final concentrations of 0.5 or 1 M to induce osmotic stress.” instead of “The response … negative control.”

Lines 184-187: “The full-length resistance [18].” delete or move to the “discussion” section.

Lines 187- use: “ For genomic DNA extraction, strains ?????????? were cultured in Potato Dextrose Broth (PDB; Beijing Aoboxing Bio-tech Co. Ltd.) at 24°C with shaking (130 rpm) for 72 h in accordance with the protocols of previous studies [18,34]. The mycelium was harvested using sterile filter paper, washed in sterile water, and flash frozen in liquid nitrogen. The total genomic DNA was then extracted using the Omega bio-tek Fungal DNA Kit (Omega bio-tek lnc., Guangzhou, China) in accordance with the protocol of the manufacturer. The resulting DNA was then used as a template to amplify the full-length sequence of the FoOs1 gene using the FoOs1-F1/FoOs1-R1 primer set (Table 2) designed with primer premier software (ver.6.0., PREMIER Biosoft, Quebec, Canada). The polymerase chain reactions (PCR) themselves, as well as the cloning and sequence analysis, were conducted using the protocol of a previous study [18]. Sequences from the fludioxonil resistant mutants and those of the sensitive parental were compared using the software DNAMAN (Lynnon Biosoft Inc. CA, USA).” instead of “The liquid cultures …. Biosoft Inc. CA, USA).”

Table 2: format as table 3 on full page.

Lines 204-206 use: “FoOs1 relative expression levels of the five laboratory fludioxonil-resistant mutants (TY-1-fR, TY-3-fR, TY-5-fR, TY-8-fR, and TG-5-fR) and their respective sensitive parental isolates (TY-1, TY-3, TY-5, TY-8, and TG-5) were assayed by quantitative real-time PCR (qPCR).” Instead of “FoOs1 (qPCR).” instead of “FoOs1 … PCR (qPCR).”

Line 207: use “as previously described, using a” instead of “previously with a”

Line 209: delete “initially”

Results sections should be arranged following the suggested materials and method subsections.

This section reports material useful for discussion

Lines 227-228: use “3.1. Generation of fludioxonil-resistant mutants

This section includes lines 229-235: No effects … the resistance was stable.

Line 235: insert: “3.2. Biological characteristics of fludioxonil-resistant mutants

3.2.a. Growth and sporulation”

This section includes lines 235- 243, figure 1 and figure 2 (lines 247-258): “Initial exploration …. isolate, TY-8.”

Figure 1: Data report the growth under different time points insert the time “0” as a 5 mm plug.

Figure 2: associate correctly graph and legend on the same page.

Lines 248-246: delete “Taken together … of resistance”

Line 259: insert “3.2.b. Cross-resistance against fungicides

This section includes lines 278-287. Use “No” instead of “The current study found no”

Line 277: insert “3.2.c. Sensitivity to osmotic stress

This section includes lines 260-267, Figure 3, and lines 269-273.

Revised lines 260-267: this section is difficult to understand

Figure 3:

As they are percentages, the maximum value is 100.

Section A and B

Use “Mycelial growth inhibition (%)” instead of “Percentage of mycelial growth inhibition (%)”

Section C and D

Explain the significance of red numbers.

Use “+100” and “-100”

As the title of the ordinate axis, I suggest:

“Percentage of mycelial growth

enhancement              inhibition”

using enhancement near the red numbers and inhibition near the black numbers

Move the acronyms of strains down at the end of the  red scale

Line 288: use “3.3.d. Cloning and sequencing analysis of the FoOs1 gene

Lines 289-291: delete. This statement is a discussion, not a result.

Lines 289- use: “Changes in the FoOs1 sequences of the fludioxonil-resistant mutants of FOM were present in Table 4.” instead of “Previous studies … Table 4).”

Table 4:

Arrange the table on the same page.

Line 313: use “3.2.e. Relative expression of FoOs1 gene

This section includes lines 322-337, Figure 4, and lines 314-325.

Supplemental information Figure 2:

Delete “CK indicates the expression level in the absence of fludioxonil.”
